

**Technical note: Ocean Alkalinity Enhancement Pelagic Impact Intercomparison Project**
**(OAEPIIP)**
Lennart T. Bach[1]*, Aaron Ferderer[1,2], Julie LaRoche[3], Kai G. Schulz[4]
[1]Institute for Marine and Antarctic Studies, University of Tasmania, TAS 7004, Australia
[2]National Collections and Marine Infrastructure, Commonwealth Scientific and Industrial
Research Organisation, Hobart, Tasmania, Australia
[3]Department of BIology Dalhousie University, Halifax, NS Canada
[4]Faculty of Science and Engineering, Southern Cross University, Lismore, NSW, Australia
*corresponding author (lennart.bach@utas.edu.au)
**Abstract**
Ocean Alkalinity Enhancement (OAE) aims to transfer carbon dioxide ($CO_2$) from the
atmosphere to the ocean by increasing the capacity of seawater to store $CO_2$. The potential
effects of OAE-induced changes in seawater chemistry on marine biology must be assessed
to understand if OAE, operated at a climate relevant scale, would be environmentally
sustainable. Here, we describe the design of the Ocean Alkalinity Enhancement Pelagic Impact
Intercomparison Project (OAEPIIP) - a standardised OAE microcosm experiment with plankton
communities to be conducted worldwide. OAEPIIP provides funding for participating
laboratories to conduct OAE experiments in their local environments. This paper constitutes
a detailed manual on the standardised methodology that shall be adopted by all OAEPIIP
participants. The individual studies will provide new insights into how plankton communities
respond to OAE. The synthesis of these standardized studies, without publication bias, will
reveal common OAE-responses that occur across geographic and environmental gradients
and are therefore particularly important to determine. The funding available to OAEPIIP and
resulting data will be shared to maximise its value and the accessibility. The globally
coordinated effort has potential to promote scientific consensus about the potential effects
of OAE on diverse plankton communities. Such consensus, through inclusion of the global



community, will provide a sounder base to facilitate political decision making whether OAE
should be upscaled or not.

**1. Rationale for the Ocean Alkalinity Enhancement Pelagic Impact Intercomparison**
**Project (OAEPIIP)**

Ocean Alkalinity Enhancement (OAE) is an emerging carbon dioxide removal (CDR) approach
(Oschlies et al., 2023). OAE drives CDR through the introduction of alkaline substances into
seawater which shift the carbonate chemistry equilibrium:

$$CO_2 + H_2O \leftrightarrow HCO_3^- + H^+ \leftrightarrow CO_3^{2-} + H^+$$

from carbon dioxide ($CO_2$) on the left to bicarbonate ($HCO_3^-$) and carbonate ions ($CO_3^{2-}$) on
the right. The decline in seawater $CO_2$ concentration lowers the seawater $CO_2$ partial pressure
($pCO_2$), thereby enabling an influx of additional atmospheric $CO_2$, or alternatively, reducing
the efflux in cases where the surface ocean is a natural source of $CO_2$ to the atmosphere. The
OAE-induced shift in carbonate chemistry is measurable as an increase in seawater alkalinity
– the name-giving feature of OAE. The viability of OAE to serve as a scalable CDR approach
critically depends on whether it is environmentally safe. Surface ocean habitats are in focus
of the environmental OAE assessment because the surface ocean is where OAE would need
to be implemented to enable $CO_2$ exchange with the atmosphere (Bach et al., 2019).
The environmental OAE assessment is only just starting but seems to be evolving in a
similar way as environmental assessments of other drivers have been set up in the past:
Research funding is provided to individual groups, who will perform individual studies in their
local environments, seeking novelty. Each of these studies will be valuable and exceeding
previous research is central to scientific progress. However, previous research on
environmental drivers has also shown that replication of experiments is perhaps equally
important as seeking novelty, since replication allows us to reveal re-occurring response
patterns across various scales and environments (Benton et al., 2007; Hamm et al., 2022;
Stewart et al., 2013). In ocean acidification research for example, an individual study found
that carbon to nitrogen (C/N) stoichiometry of plankton communities is increased under high
$CO_2$ conditions due to $CO_2$ fertilization of the phytoplankton community (Riebesell et al.,



2007). However, replication of the experiment at different locations found that zooplankton
communities can strongly modify the response, to the point that the response can be
significant in the opposite direction (lower C/N under high $CO_2$ (Taucher et al., 2021)).
Arguably, the crucial progress in this example was understanding of the context-dependency
of the C/N response to ocean acidification, which was made possible by replication of a
sophisticated experiment across a wide geographical range (Riebesell et al., 2013). Likewise,
the intercomparison of climate models via replicated numerical experiments (Dingley et al.,
2023) has long been recognised as a cornerstone to the assessment of climate change
(Masson-Delmotte et al., 2021), possibly more influential than the output of individual
climate models.
The Ocean Alkalinity Enhancement Pelagic Impact Intercomparison Project (OAEPIIP) builds
upon these insights from previous environmental assessments by establishing a platform that
supports replication, while still enabling the pursuit of novelty. In essence, OAEPIIP provides
funding for a cost-efficient and standardised OAE experiment, which can be conducted by
scientists across the globe (section 2). The experiments will use a microcosm setup to study
the response of natural plankton communities to two specific OAE scenarios, and they will
determine the same set of response variables. Each experiment shall be published on an
individual basis in a special issue of a peer-reviewed scientific journal under open access with
costs largely covered by OAEPIIP (section 3). Individual publication of OAEPIIP experiments
gives room to describe novel observations on how plankton communities respond to OAE. All
datasets will be shared and synthesized in a meta-analysis.  The standardised experimental
design facilitates inclusion of individual datasets into the meta-analysis (Harrison, 2011).
Likewise, the collection of all datasets, irrespective of their outcomes, avoids publication bias,
which is a known problem of meta-analyses (Field and Gillett, 2010). We expect OAEPIIP to
promote consensus among scientists concerning the potential environmental side effects of
OAE on plankton communities, with significant potential for capacity building (section 4). This
paper provides a detailed manual for the OAEPIIP experimental setup and describes its
benefits.

**2.** **Experimental infrastructure, operation, and design**

**2.1. Microcosm setup**




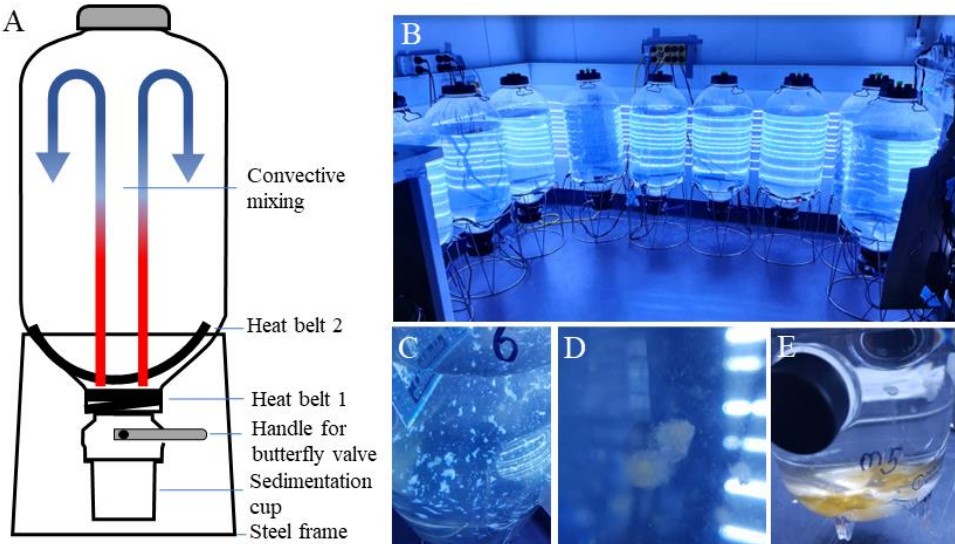


**Figure 1.** *Microcosm setup. (A) Schematic of the microcosm tanks. The 2 heat belts induce convective mixing within the tanks. (B) Arrangement of 9 microcosms in a temperature-controlled room in front of a light source. Their position should be changed on a daily basis to minimize position-dependent differences in light and temperature over the course of the study. (C) A picture of a microcosm, just after NaOH addition. The white flakes are brucite particles that need to be dissolved after NaOH addition by stirring the seawater within microcosms with a plastic spoon. (D) A close-up of a marine snow aggregate, which frequently forms after a phytoplankton bloom. (E) Marine snow aggregates collected in the sedimentation cup of the microcosm. Sampling these can be interesting, although this is not an essential parameter of OAEPIIP (section 2.6.).*

OAEPIIP utilizes the microcosm setup developed by Ferderer et al. (2022), as it is cost-effective and relatively easy to set up and operate. The microcosms are 55L Polyethylene terephthalate (PET) tanks, which were originally designed for home brewing (Fig. 1). The tanks are mounted on steel frames and have 120 and 70 mm openings at the top and bottom, respectively. The bottom opening is equipped with a butterfly valve and a sedimentation cup, used for the collection of settling material. The butterfly valve has a handle so that the sedimentation cup can be isolated from the water column.



The crucial steps for setting up the microcosms, their filling, and their operation are listed in
Table S1, illustrated in Figs. 1 and 2 and Videos S1 (https://doi.org/10.5446/66751)  and S2
(https://doi.org/10.5446/66753), and briefly described here. OAEPIIP experiments occupy
approximately 9 m$^2$ in a temperature-controlled room with a cooling capacity of roughly 6°C
below the temperature aimed for in the microcosm study (e.g., to 14°C if the desired
experimental water temperature is 20°C). The microcosms need to be thoroughly cleaned
before use (Table S1).
Infrastructure needed for filling the microcosms with natural seawater (containing natural
plankton communities) depends on the local environment at an OAEPIIP study site. At our
site in Tasmania, we fill microcosms from a jetty using a small crane or davit (Fig. 2; Video S1
(https://doi.org/10.5446/66751)). Natural seawater with plankton communities shall be
collected by opening the top lid and butterfly valve at the bottom and lowering the
microcosms slowly into seawater so that each microcosm is filled from bottom to top. Care
must be taken to not enclose larger debris, nekton or sediments. Once the microcosm is
submersed and only the upper opening is above the sea surface, a rope attached to the
handle of the butterfly valve is pulled so that the bottom opening is closed. The microcosm
can now be lifted back on shore and put back into its metal frame. Another possibility to fill
microcosms is to slowly lower them from a low swimming pontoon or small boat and close
the bottom manually. Filling microcosms by slowly lowering them into seawater is a very
gentle way to collect plankton communities (Video S1 (https://doi.org/10.5446/66751)),
avoiding the physical disturbance to plankton imposed by pumping. Based on our experience
it takes roughly 45 minutes to fill 9 microcosms. Longer timescales for the collection (i.e. >>1
hour) should be avoided to mitigate the risk of changes in seawater communities over the
course of the filling procedure (e.g. through tidal water movement). This potential problem
should also be minimised by filling the microcosms in random order. Furthermore, care
should be taken to not expose the microcosms to excessive sunlight (or heat) after filling.
The weight of the enclosed seawater needs to be determined after the filling procedure as
this information is needed for establishing treatments (section 2.5). This could be done using
a balance or (if a balance is not available) volumetrically and determining weight with known
volume, temperature, and salinity. Once the weight has been determined, microcosms need
to be transported to the temperature-controlled room where the experiment takes place and
light and temperature control needs to be initiated immediately (see following section).




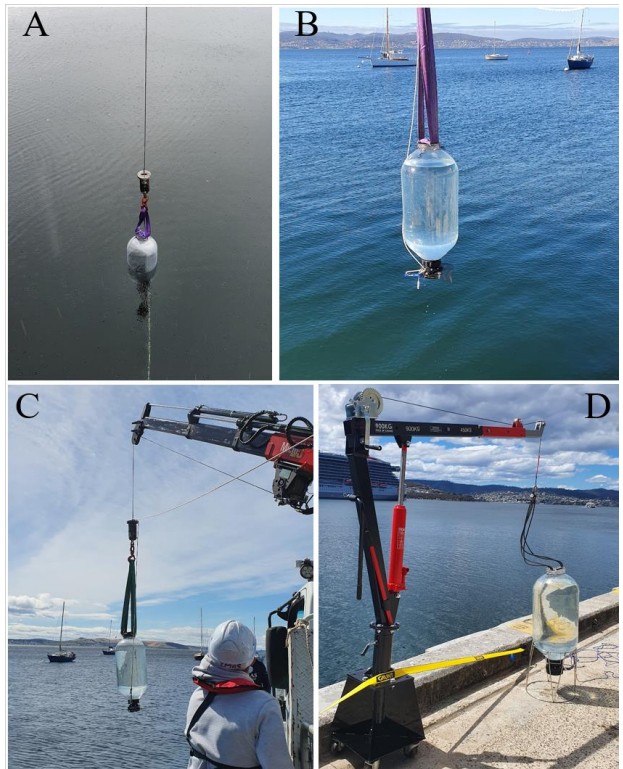


***Figure 2****. Seawater collection for the microcosm experiments. (A) A microcosm slowly lowered*
*into seawater to gently collect a plankton community. (B and C) A filled microcosm being*
*pulled back on land. Please note that we mostly used a small crane mounted to a truck or a*
*davit (as in D) for the seawater collection. However, microcosms filled with seawater only*
*weigh about 60 kg, so that lighter gear is probably sufficient for collection. A detailed*
*description of seawater collection is provided in Table S1 and Video S1*
*([https://doi.org/10.5446/66751](https://doi.org/10.5446/66751)).*

**2.2. Mixing, temperature, light, and nutrient conditions in OAEPIIP experiments**

OAEPIIP utilises convection to mix the enclosed microcosm volume and keep plankton in
suspension (Fig. 1). To establish convective mixing, two 30 Watts heat belts will be firmly
attached to two distinct locations at the bottom of the microcosms (Fig. 1, Video S2
([https://doi.org/10.5446/66753](https://doi.org/10.5446/66753))). Based on our experience, these heat belts increase the



temperature of the enclosed seawater by ~6°C relative to the room temperature, so that
room temperature needs to be roughly ~6°C lower than the target temperature in the
experiments (please note that testing the temperature difference will be necessary prior to
the experiment as temperature offset may differ across temperature-controlled rooms).
Once heat belts are attached, microcosms should be placed in front of the light source and
heat belts should be plugged in to initiate the convective mixing.
While convection provides gentle and non-invasive mixing, there are several trade-offs in
regard to temperature control. Firstly, due to the removal of seawater during sampling, the
total volume within microcosms declines over the course of the experiment. Since the heat
belts cannot be adjusted, there is an increase in heat energy input per liter of enclosed
seawater and thus a gradual warming. To mitigate this issue,  the external cooling may need
to be increased over time by lowering the room temperature. In our experience, a reduction
by 1°C for every 5 liters of seawater sampled from the microcosms is sufficient to keep the
seawater temperature relatively constant over the course of the study. Secondly, small
differences in ventilation at different locations in the temperature-controlled room can lead
to seawater temperature differences of around 2°C between microcosms (Ferderer et al.,
2022; Guo et al., 2023). To mitigate this experimental constraint, the microcosm placement
within the temperature-controlled room must be shuffled on a daily basis. Microcosms can
easily be moved when they are being pulled on the steel frame (Fig. 1), but care must be taken
to briefly unplug the heat belts and plug them in again after shuffling their position.
Furthermore, fans can be utilised to remove heat pockets in the room, although care must be
taken as the wind can have a strong cooling effect, resulting in a microcosm that was too
warm quickly becoming too cold. Since temperature is a strong driver of physiological
processes, it is highly advisable to thoroughly test the setup with all microcosms prior to the
experiment. The goal should be to have as little variation in temperatures between
microcosms as possible, and the temperature should be as the plankton community would
have experienced it at the location/season it was collected.
Like temperature, light conditions reproduce the natural site-specific conditions as much as
possible. This applies for the light/dark cycle, the light intensity, and the light spectrum (light
spectrum should be between 400 and 750 nm, i.e. cool white light). Since many OAEPIIP
participants may not have access to sophisticated computer-controlled light sources, we
recommend the delivery of constant light over a fixed light/dark cycle. In an OAE study in



Tasmania, for example, we provided light constantly with 200 µmol photons m$^{-2}$ s$^{-1}$ at a 12/12
hours light/dark cycle. These conditions were considered as representative average level for
the surface mixed layer at the location/season where/when the natural plankton community
were sourced in Tasmania. The light/dark cycle can be achieved by plugging the light source
into a timer socket. The microcosms need to be positioned in such a way that light is very
similar inside each microcosm. A light meter shall be used to determine light intensity inside
the microcosms prior to the experiments and at the end (Table 1). Positioning can be very
critical since movement by a few centimetres can often lead to noticeable changes in
measured light intensity that are undetectable by the human eye. It is therefore important to
adjust light conditions before starting the experiment and marking the spot on the floor
where individual microcosms must be placed to ensure replicable light levels. It is also
important to have all the other microcosms at their respective locations while doing the
adjustments as they might shade each other. The daily shuffling of microcosm positions inside
the room, which is essential for the temperature control (see above), will also help to mitigate
systematic bias in light regime between microcosms.
OAEPIIP experiments shall not add organisms, nutrients, or any substances other than
alkalinity/DIC (section 2.4) to the microcosms during or before the experiments.

**2.3. OAE method in focus of OAEPIIP**

OAE can be implemented with different approaches (Eisaman et al., 2023), which have
different environmental implications as they are associated with different environmental
perturbations (Bach et al., 2019). A widely considered approach is electrodialytical OAE,
where liquid sodium hydroxide (NaOH) is the alkalinity source (de Lannoy et al., 2018). NaOH-
based OAE is in focus of OAEPIIP due to the following reasons. First, electrodialytical OAE was
recently evaluated to rank among the highest OAE approaches with regards to their
"technological readiness level" (Eisaman et al., 2023), with field trials already underway.
Second, liquid NaOH is suitable as an alkalinity source for applications in pelagic environments
as it delivers quasi instantaneous OAE in seawater. Other methods that involve more slowly
dissolving minerals (e.g. olivine) are considered less suitable for pelagic applications as they
would partially sink into the deep ocean before dissolving (Köhler et al., 2013; Fakhraee et al.,
2023). Third, electrodialytical OAE is chemically relatively similar to other OAE methods such





as OAE with magnesium hydroxides or ocean liming based on calcium hydroxides. Like NaOH,
magnesium and calcium hydroxides dissolve relatively quickly and are comparatively clean
sources of alkalinity so that their primary potential to affect pelagic communities is by
changing seawater carbonate chemistry. Thus, results from NaOH-based OAE experiments
will also inform these other approaches. Fourth, NaOH is readily available worldwide, which
is logistically beneficial for OAEPIIP.

**2.4. Experimental design**

NaOH-based OAE reduces seawater $pCO_2$ within seconds (Zeebe and Wolf-Gladrow, 2001),
whereas the subsequent equilibration with atmospheric $CO_2$ takes months to years (Jones et
al., 2014) or even longer (He and Tyka, 2023). The carbonate chemistry perturbation is much
greater before the equilibration has happened so that more pronounced effects on
communities would be expected shortly after alkalinity addition (Bach et al., 2019). As such,
an argument can be made to study OAE in two different scenarios when using rapidly
dissolving alkalinity sources like NaOH or other hydroxides. These are the "unequilibrated"
scenario, simulating the fact that $CO_2$ influx has not yet happened right after alkalinity
addition, and the "equilibrated" scenario, assuming the alkalinity enhanced seawater has
already $CO_2$-equilibrated with the atmosphere.
The nine microcosms available for OAEPIIP experiments will provide triplicate incubations for
controls, unequilibrated and equilibrated treatments. An important aspect for OAEPIIP
experiments is that the amount of alkalinity added to the treatments is consistent among all
studies. Modelling studies suggest that gigatonne-scale OAE sustained for 80 years would
increase surface ocean alkalinity by about 100-200 µmol/kg (Burt et al., 2021; Lenton et al.,
2018). This seemingly modest perturbation is due to dilution by the huge volume of the ocean
(i.e., $9.44*10^{17}$ m³; Sarmiento and Gruber, 2006). However, the perturbation can be more
pronounced at sites where alkalinity is added, before being diluted with unperturbed
seawater (He and Tyka, 2023). Based on this, we determined an addition of 500 µmol/kg to
both the unequilibrated and equilibrated treatments for OAEPIIP. While a 500 µmol/kg
alkalinity increase is on the higher end for what is plausible for OAE, it seems to be a good
compromise between realism and the ability to detect environmental effects on plankton
communities (Ferderer et al., 2022).




### 2.5. Establishing treatments


Alkalinity enhancement shall be performed on day 0 of the experiment, shortly after
microcosms have been positioned in the temperature-controlled room. Before adding
alkalinity, carbonate chemistry samples (i.e., alkalinity and one other carbonate chemistry
parameter; section 2.6) should be collected to constrain carbonate chemistry conditions in all
microcosms before OAE.
The three control microcosms will not receive any alkalinity addition and remain untreated.
The three microcosms of the unequilibrated treatment will receive 500 µmol/kg of NaOH. The
simplest way to achieve this is by purchasing and using a 1 molar NaOH solution (ideally in
"analytical quality") and adding 500 µL per kg of enclosed seawater. For example, if 54.5 kg
of seawater have been enclosed then 54.5 * 500 = 27250 µL of 1 molar NaOH solution needs
to be added to the respective microcosm.
The equilibrated treatment is slightly more complicated to establish. Here, most of the
alkalinity needs to be added as sodium bicarbonate ($NaHCO_3$) solution and a smaller amount
as NaOH solution. We provide an R script based on Seacarb (Gattuso et al., 2021) that can be
used to calculate additions of $NaHCO_3$ and NaOH (OAEPIIP, 2024). Furthermore, video
tutorials provide detailed instructions on how to use the R script or how to do these
calculations with CO2SYS for MSExcel (Pierrot et al., 2021) (Videos S3 and S4;
https://doi.org/10.5446/66754, https://doi.org/10.5446/66752). Briefly: In a first step, initial
carbonate chemistry conditions need to be calculated for the unperturbed seawater enclosed
in the microcosms. For this calculation one needs to assume a current $CO_2$ partial pressure
(e.g., 420 µatm), the target temperature for the experiment, and a salinity and alkalinity
estimate based on what the experimentalist expects for their region (or ideally has measured
just before collecting the seawater for microcosm experiment). Next, the calculation is
repeated for the same conditions except for alkalinity where 500 µmol/kg is added to the
assumed value (e.g., 2850 µmol/kg when the assumed value of the unperturbed water was
2350 µmol/kg). The second calculation represents the desired conditions in the equilibrated
treatment after the alkalinity enhancement. The calculated dissolved inorganic carbon (DIC)
concentrations of the initial carbonate system ($DIC_{initial}$) need to be subtracted from the
calculated DIC of the calculated treatment ($DIC_{equilibrated}$):




$$NaHCO_3\ addition =\ DIC_{equilibrated} - DIC_{initial}$$



where NaHCO$_3$ addition is the amount of NaHCO$_3$ that needs to be added per kg of enclosed
seawater (in µmol/kg). The addition of NaHCO$_3$ provides equal amounts of DIC and alkalinity.
However, OAE can only absorb ~0.85 mole of DIC per mole of alkalinity added (He and Tyka,
2023; Schulz et al., 2023), so that reaching to +500 µmol/kg requires the addition of slightly
more alkalinity without DIC. NaOH is used for this purpose and the exact amount that needs
to be added is calculated as:


$$NaOH\ addition = 500 -\ NaHCO_3\ addition$$


Where 500 is the targeted alkalinity enhancement in µmol/kg. NaHCO$_3$ and NaOH additions
need to be multiplied with the weight of the enclosed microcosm seawater to calculate how
much NaHCO$_3$ and NaOH need to be added per individual microcosm.
It is recommended to use 1 molar stock solutions for both NaHCO$_3$ and NaOH for treatment
manipulations because in that case required additions in µmol/microcosm are equivalent to
µL/microcosm. For example, in the equilibrated treatment a typical addition would be 420
µL/kg of NaHCO$_3$ and 80 µL/kg of NaOH (i.e., 22.89 mL/microcosm NaHCO$_3$ and 4.36
mL/microcosm NaOH when 54.5 kg of seawater were enclosed). One molar NaHCO$_3$ stock
solutions can be prepared by dissolving 8.4 g NaHCO$_3$ powder (dried at 60°C overnight ; note
that NaHCO$_3$ decomposes at higher temperatures) in 100 mL deionised water. One molar
NaOH (ideally in "analytical quality") should be purchased as such.
The addition of NaOH to seawater causes precipitation of magnesium hydroxides, which
appear as white flakes (Fig. 1C). Therefore, microcosms should be stirred with a clean plastic
paddle during and after NaOH additions until all white flakes disappear. This problem will be
particularly pronounced in the unequilibrated treatment where all alkalinity is added as
NaOH. For consistency, control and equilibrated microcosms should be stirred as much as the
unequilibrated microcosms. If OAEPIIP participants do not have prior practical experience
with seawater carbonate chemistry manipulation, it is advised to test the above mentioned
procedures (including the measurement of resulting carbonate chemistry parameter changes
such as in TA and DIC) before commencing the main OAEPIIP experiment.






**2.6. Essential parameters to be measured in OAEPIIP experiments**


Next to an identical experimental design and setup, the same parameters need to be measured in individual OAEPIIP experiments to make them comparable. A list of "core" parameters with justifications for their choice is provided in Table 1, and additional recommendations on how to sample and process these is provided in Table S2. The core parameters (Table 1) should provide a relatively comprehensive, yet cost-efficient insight into processes within the plankton community. Although all core parameters need to be measured in all participating OAEPIIP studies, there may be unsurmountable logistical constraints which prohibit a participant from determining a core parameter. Such cases should be mentioned upon application for OAEPIIP participation so that mitigation pathways can be explored and that potential participants with less infrastructure capacity still have the opportunity to participate if possible (see also section 4).

If they wish to do so, OAEPIIP participants can also measure additional parameters to maximise their individual experimental outcomes. However, the following issues should be considered:

341

1) Not more than approximately 1/3 of the microcosm volume should be sampled over the course of the study to avoid too much heat input per liter of enclosed volume via the heat belts (the room temperature might need lowering to compensate for reducing volume throughout the experiment; section 2.2).

2) Any type of contamination (particulate or dissolved organic or inorganic) must be kept at a minimum.

3) It is possible to sample mesozooplankton with a customized net (Guo et al., 2023), but sampling should be restricted to 3 occasions during the experiment (e.g., beginning, middle, end) to avoid overfishing.

4) Aggregation and sedimentation are often observed in these microcosm studies and it is encouraged to sample sedimenting materials from the sediment trap (Ferderer et al., 2022). However, care must be taken to not remove significant volumes of seawater.




**Table 1.** *List of core parameters that essentially need to be measured in all individual OAEPIIP*
*studies. The "Samplings" column indicates how often all 9 microcosms need to be sampled for*
*a specific parameter during the study. "Daily" means that this parameter needs to be*
*measured every day, irrespective of the temperature-dependent duration of the study. "b/e"*
*means that samples need to be taken at the beginning and the end of the experiment.*

| Core parameter | Rationale | Samplings |
|---|---|---|
| Alkalinity | The treatment-defining parameter of the study. | 7* |
| Second carbonate chemistry parameter (e.g., pH or DIC) | Required to constrain the carbonate system. Also provides insights for net autotrophy/heterotrophy. | Daily* |
| Salinity | Required to define the marine system under investigation. | b/e |
| Light | To constrain physical conditions for growth. | b/e |
| Temperature | To monitor its influence on metabolic rates and assess temperature stability due to convective mixing. | daily |
| Nutrients ($NO_x^-$, $PO_4^{3-}$, $Si(OH)_4$) | Nitrate+Nitrite ($NO_x^-$) and phosphate ($PO_4^{3-}$) availability largely determines the productivity of the plankton community. Availability of $Si(OH)_4$ provides insights if productivity will likely be driven by diatoms. | 11 |
| Chlorophyll a (chla) | Chla is a widely used proxy for phytoplankton biomass | 11 |
| Particulate organic carbon and nitrogen (POC and PON) | POC and PON dynamics are related to the increase and decline of biomass. Their ratio (POC/PON) is an important metric in biogeochemical element cycling. | 11 |
| Biogenic silica (BSi) | BSi is a widely used proxy for diatom biomass | 11 |
| Flow cytometry (FC) | FC is a cost-efficient tool that reveals shifts in phytoplankton size classes and specific groups with distinguishable fluorescence/scatter characteristics. FC is particularly good for enumeration of small phytoplankton and heterotrophic bacteria. | 11 |
| Microscopy | Microscopy is a widely available tool to assess dynamics in phytoplankton and microzooplankton communities. It is complementary to FC as it is better suited for larger phytoplankton/microzooplankton. | 7 |
| Nucleic acid sample | Nucleic acid samples (DNA and possibly RNA) will provide a detailed assessment of microbial diversity. Basic requirements for this parameter will be metabarcoding for 16S rRNA genes (variable region of V4-V5). Further analysis for metagenomics and metatranscriptomics will be possible depending on the timing of sample collection but are not essential for the participation. | b/e |

*\*These parameters must be sampled directly before and after establishment of the OAE*
*treatments in all 9 microcosms. All other parameters must be sampled for the first time after*
*establishment of the treatments.*




**2.7. Duration of experiment**


To the best of our knowledge, there is no general rule for the ideal duration of microcosm
experiments. Experiments that are too short may miss important responses of plankton
communities while long experiments may exacerbate so-called "bottle effects", non-specific
effects from confinement rather than the experimental perturbation itself (Pernthaler and
Amann, 2005). Based on experiments with the OAEPIIP setup in Tasmania we consider 20
days as a good compromise for an experiment at 15°C. However, metabolic rates increase
with temperature so that experimental duration needs to be adjusted based on respective
locations. Informed by Q10 temperature dependencies (Sherman et al., 2016), we
recommend the following framework: 20 days is the reference duration at 15°C. The duration
(in days) increases/decreases from this reference point using Q10 kinetics:

$$Duration = \frac{0.5611}{\left(0.5611 \times 1.47^{\frac{T_{exp}-15}{10}}\right)} \times 20$$


where 0.5611 is the reference growth rate at 15°C, $T_{exp}$ is the anticipated temperature in the
OAEPIIP experiment, and 1.47 is the $Q_{10}$ factor derived by (Sherman et al., 2016). For example,
an experiment at 25°C should last for 14 days and an experiment at 5°C for 29 days.

**2.8. Sampling operations and logistics**

The convective system mixes the water column so that no manual mixing is needed prior to
sampling. A peristaltic pump is recommended to withdraw the seawater samples from the
microcosms.
The total number of samplings for specific parameters is listed in Table 1 (for example, POC
and PON need to be sampled 11 times in total). The frequency of sampling needs to be
adjusted based on the temperature-dependent duration of the experiment (section 2.7.).
OAEPIIP experiments at higher temperatures require higher sampling frequency because
metabolic processes are faster. Table 1 lists the minimum number of days a parameter should
be sampled. This number is to guarantee that there will be enough comparable data points



across OAEPIIP experiments. For example, nutrient samples should be taken at least 11 times
in each microcosm during the experiment. For an experiment at 15°C (20 days), this could
mean a sampling on day 0 (directly after establishing treatments) and then days 2, 4, 6,…,20.
However it may also be reasonable to increase frequency during periods of phytoplankton
blooms (e.g., daily) and then reduce the frequency (e.g. every 4 days) when nutrients are
depleted. In general, OAEPIIP experimentalists can best decide on an individual basis what
sampling schedule is most appropriate for their experiment, but the total number of
samplings must be at least as defined in Table 1 for each of the listed parameters.
Sampling for all OAEPIIP experiments should begin two hours after the onset of the light
period on a sampling day. This coordination of initial sampling ensures that the plankton
community is in a similar diurnal growth state. Hence, sampling of all 9 microcosms should
ideally not last longer than 3 hours.

**2.9. Statistical analyses**

Microcosm data contains complex ecological data which require specific (often complicated)
statistical tools for their analysis. A common issue is the presence of non linear relationships,
which without gross transformation of the variables prevents the fitting of data to linear
models. Furthermore, OAEPIIP microcosms will be sampled several times over an extended
period. This sampling strategy results in temporal-pseudoreplication, where observations are
not independent of each other and therefore violate the assumption of independence
required for simple linear models and Generalised additive models (GAMs). The expansion of
GAMs to Generalized Additive Mixed Models (GAMMs) allows for correlations between
observations and the modelling of data structures which are nested as well as for non-linear
relationships between the response and explanatory variables.
To facilitate and standardize statistical analyses of individual datasets we provide an R-based
pipeline (OAEPIIP, 2024). This pipeline is tailored towards the evaluation of individual OAEPIIP
data sets using GAMMs. The files contain a workflow which demonstrates the use of GAMMs
and facilitates the seamless integration of individual datasets gathered during OAEPIIP
experiments into the workflow. Theoretical background, knowledge and details on how to fit
such models can be found in the textbooks by Zuur et al. (2009) and Wood (2017).



**3. Logistics and administration**

Basic instructions and updates on OAEPIIP will be provided on the OAEPIIP website
([https://appliedbgc.imas.utas.edu.au/ocean-alkalinity-enhancement-pelagic-impact-](https://appliedbgc.imas.utas.edu.au/ocean-alkalinity-enhancement-pelagic-impact-)
[intercomparison-project/](https://appliedbgc.imas.utas.edu.au/ocean-alkalinity-enhancement-pelagic-impact-intercomparison-project/)).

**3.1. Eligibility and funding**

To join OAEPIIP, participants need to be capable of performing an OAEPIIP study and provide
all data by December 2025. This capacity shall be confirmed on a simple 1 page form (available
on the OAEPIIP website) that potential participants need to fill in and send to the email
provided on the form. Career stage, publication record, or other parameters of a scientist's
curriculum vitae have no relevance for OAEPIIP. As such, application success is determined by
logistical and infrastructure-related aspects, for example whether a participant has access to
a temperature-controlled room and can provide the various data in the given timeframe (but
see also section 4 on suggestions on how to mitigate individual limitations to infrastructure).
Ultimately, participation is restricted by total funding available to OAEPIIP. Should there be
more applications than there is funding, participants will be selected based on two criteria:
First, we will consider the locations of the experiments to obtain the best possible geographic
spread. Second, participants will be selected by chance should there be clusters of
applications in close proximity.
OAEPIIP provides a maximum of around 12,000 US$ per study in materials and funding for
analytical costs and publication fees (the exact amount is slightly variable due to exchange
rates). Standardized components like the microcosms will be supplied and fees for the
publication of individual studies in an OAEPIIP special issue will be covered (see section 3.2).
The remaining funding will be transferred via invoicing. Thus, participants must have a bank
account associated with their affiliation to which funding can be transferred from Australia.
This criterion therefore excludes laboratories in countries under relevant sanctions from
Australia to receive funding, although they are still welcome to be part of the OAEPIIP
community. Practically, participants will send two invoices to the University of Tasmania, one
at the beginning of the experiment to support purchasing of materials (e.g. the microcosms)
and the second one towards the end when the data is available and has been submitted.



OAEPIIP cannot provide funding for salaries. Therefore, the experiment was designed to be
suitable for a Master thesis or a chapter of a PhD thesis.

462         **3.2. Data management and publication**


Datasets of individual OAEPIIP studies should be formatted using a standardised template
available on the OAEPIIP homepage (section 3) and submitted to OAEPIIP as soon as they are
available. All data must be uploaded and made available under open access. Participants will
be listed on the OAEPIIP homepage and their individual datasets will be linked to their names
and affiliations as soon as it is made available. OAEPIIP experiments shall be published on an
individual basis in an OAEPIIP special issue (publication fees of up to 1600 US$ are part of the
~12,000 US$ funding provided by OAEPIIP). Individual publication will enable identification of
novel observations on how plankton communities respond to OAE. If participants prefer not
to publish their data they still need to submit their data to OAEPIIP so that it can be included
in the OAEPIIP synthesis. This is critically important because the synthesis must avoid
publication bias.
The OAEPIIP synthesis will be prepared once all datasets have been delivered. First and last
authors of individual studies will automatically be co-authors on the synthesis publication(s)
at the end of the project, unless they prefer not to be.

479         **4. Capacity building and inclusivity**


OAEPIIP has potential benefits that go beyond scientific knowledge gain. The community
effort helps to build a network of OAE scientists and provides an incentive and access to those
who have not yet engaged with OAE research. Indeed, growing the OAE research community
is essential to accelerate the OAE assessment and make it more comprehensive. Providing
the same amount of funding, regardless of the location, may increase the attractiveness of
OAEPIIP studies to those that currently have less funding. Participation of scientists
worldwide is what we aim for since the OAE assessment requires the inclusion of the global
community. Indeed, participation in the process of assessing marine CDR methods (such as
OAE), rather than being on the receiving end of information only, has been expressed as an
important aspect by stakeholders from developing countries.





We are aware that the infrastructure demands for OAEPIIP (section 2), still put barriers on
participation. To mitigate those barriers, potential participants from more experienced
laboratories can offer to serve as a partner for a less experienced laboratory. Likewise,
potential participants from less experienced laboratories can indicate if they essentially need
support from an experienced laboratory. This information shall be disclosed on the
application form (available on the OAEPIIP website) so that OAEPIIP can establish
partnerships between participants. Partners can support each other through knowledge
exchange but also more practically by analysing samples for each other. For example, if an
interested participant has no capacity to measure alkalinity or flow cytometry samples, it may
partner with another participant to share analytical duties. The distribution of funding for
analytical costs via invoicing allows for such flexibility as it provides an opportunity to easily
re-distribute funding between project participants when this is communicated with the
OAEPIIP administration. For example, when two laboratories partner, they together have
access to twice the funding  (~24,000 US\$), which they share among them for the two
experiments they would have to do (the two experiments must be at different locations to
guarantee geographical diversity).
Furthermore, potential participants that simply have no chance to measure one (or more)
core parameters due to unsurmountable logistical constraints can still hand in an application,
if they indicate which parameters they are unable to deliver on their application form
(available on the OAEPIIP website). The OAEPIIP administration will then evaluate such
applications on a case-by-case basis and explore if there is a way for participation despite this
limitation. This pathway is set in place specifically for potential participants with less
developed infrastructure and less capacity for collaboration with an experienced (e.g., due to
geographic isolation).
Altogether, we hope the cost-efficient design of OAEPIIP, its eligibility criteria that refrain
from classic measures of scientific success, and potential support via an evolving OAEPIIP
community could promote an inclusive assessment of OAE. One primary goal of OAEPIIP is
capacity building to provide more informed decisions concerning OAE that encompass data
from a geographically diverse range of plankton ecosystems.

**Competing interests**



The contact author has declared that none of the authors has any competing interests.

**Acknowledgements**

We thank the Carbon-to-Sea Initiative for funding OAEPIIP. LTB acknowledges funding from the Australian Research Council by Future Fellowship (FT200100846).

**Data availability**

All code provided for experimental design and statistical analysis can be found here: (OAEPIIP, 2024).

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
