# Peer review of "Technical note: Ocean Alkalinity Enhancement Pelagic Impact Intercomparison Project"

_EGUsphere, 2024_

## Author Response (AR1)

Comment by Editor Dr. Patricia Grasse:

When performing OAE experiments with NaOH, participants should be aware that the formation of brucite (Mg(OH)2) at higher pH (approximately >9), also precipitates phosphate (Karl and Thien, 1992, Table 1), silicate, and carbonates. I am not sure if you have seen this in previous experiments. The formation of brucite depends on the pH, and the amount of Mg (and possibly Ca) in the natural seawater. I think this is something people should be aware of when doing these experiments. Maybe you can add the information in 2.5 or somewhere else. However, as long as the pH does not exceed 9 (9.5) and the "flakes" dissolve again, this is not an issue.

REPLY: Thank you, excellent point. We addressed this issue in section 2.5 to make the reader aware of the phosphate problem and how to avoid it. In our previous studies with the OAEPIIP treatment design (one published by Ferderer et al., (2022)), we had identical phosphate concentrations after establishing treatments, confirming your statement that dissolution of brucite solves the problem.

Reviewer #1

Comment 1:

This technical note describes a proposed intercomparison project in which standardized microcosm experiments will allow for study of natural plankton communities to two specific ocean alkalinity enhancement scenarios. Collection of the same response variables will allow for meta-analysis to improve understanding of potential environmental side effects of OAE on plankton communities. The methods described in this note, and the need for comparable OAE studies, are highly relevant to the scope of this journal and to advancing the field of marine carbon dioxide removal broadly. I recommend publication with minor revisions, as noted in specific comments below, with the caveat that this note could be significantly strengthened with an overview of recent studies on the environmental side effects of OAE, to place the suggested methods in context of similar work. Finally, these studies will all be completed in microcosms and there is limited discussion of extension to field experiments. This would be helpful context for those interested in participating and in using the eventual results for both field and modelling studies.

REPLY: We thank the Reviewer for their support and the time and effort of reviewing our manuscript. The general comment provided above will be addressed in the specific comments below. Please note that line numbers we are referencing below refer to the revised version of the manuscript.

Specific comments:

Comment 2

Paragraph 1 (Line 38-52) could use some additional references on OAE, particularly the NASEM 2021 report. A brief description of the different types of alkalinity sources would be useful as upfront context for interdisciplinary audiences.

REPLY: Thank you, we added the NASEM reference to the first paragraph and section 2.3. We also added additional detail on the variety of alkaline materials that can potentially be used for OAE in section 2.3. We refer to (Eisaman et al., 2023) here as these authors have just provided a comprehensive assessment of the different OAE implementation strategies.

Comment 3

Line 54: Environmental drivers are referenced several times in this section but with limited explanation—drivers of CO2 exchange? Of environmental change, or climate change? Specificity with strengthen this section, especially as the examples range from ocean acidification research to climate models.

REPLY: Thank you, good point. To clarify we specified the term "driver" here.

Comment 4

Line 79: Two OAE scenarios leads me to believe that this considers 2 different alkalinity feedstocks, methods of alkalinity delivery, or significantly varying concentrations of alkalinity. Later sections highlight that this is in fact 1 control, 1 NaOH-based treatment (just after alkalinity dosing), and 1 equilibrated treatment (after air-sea exchange of CO2). This should be clarified upfront, that this is not two OAE scenarios, but essentially 2 timepoints in OAE—right after delivery and after equilibration.

REPLY: Agreed, scenarios is not an ideal term to describe this in this part of the text. We changed the description and refer to "one widely considered OAE implementation strategy" on line 79. When the experimental design is described in detail we will refer to two different timepoints after the alkalinity input. We further added a sentence how equilibration could be achieved (i.e., over time or by enforcing it in a facility).

Comment 5

Line 88: This would be a good place for a comprehensive discussion of the potential environmental side effects under consideration, which aren't touched on much throughout this manuscript.

REPLY: Thank you for this comment. While we agree that a comprehensive environmental assessment may be useful in other contexts, we refrain from this here because our goal is to keep the assessment fully neutral. If we already provided hypotheses what OAEPIIP participants may be looking for, we would introduce a certain degree of conscious or subconscious bias.

Comment 6

Line 89: capacity building of what? Of OAE? Of microcosm experiments? Of personnel and infrastructure?

REPLY: The specific capacity building benefits we would anticipate are described in section 4. The section referred to by the Reviewer here is mainly as an overview and to point the reader to the sections where further detail can be found.

Comment 7

Figure 1: Some sense of scale here would be appreciated. Are there any sensors in these tanks? Are the microcosms sealed from the atmosphere? This comes across in the text but would be strengthened by inclusion in the figure.

REPLY: Thank you, we have added a scale bar to the figure and indicated the enclosed headspace. With regards to sensors, we don't consider permanently installed sensors in the standard setup, but participants may still chose adding sensors if it helps their individual experiment and does not interfere with the experimental design. It is hard to explain this in a drawing and therefore refer to the text for this specific aspect.

Comment 8

Line 110: Source of tanks? Part numbers? Are frames custom ordered? If yes, what are the details? I would expect a manual for microcosm setup to include specific details for replication without requiring the reader to dig through Ferderer 2022, in case the setup is not exactly the same. Later sections note that at least some of this equipment is available to participants—can non-funded labs wanting to replicate the methods also purchase this equipment from this group? If not, they need specific direction to replicate the experiment.

REPLY: Thank you, excellent points. We have provided additional detail on the specifics of the "FermZilla" microcosm incubators (and heat belts) in Table S1 and also emphasized in the main text that all participants need to use these FermZilla tanks to establish comparability between OAEPIIP experiments. We changed the latter section because with the first OAEPIIP participants which are already commencing their work, we realized that it is logistically easier to provide the money and they purchase the FermZilla tanks themselves. With these amendments to the text, it is also clear where other labs (who want to do same or similar research) can purchase the equipment.

Comment 9

Line 121: How are the microcosms cleaned? Soap and water? Acid? DI H2O? Useful details for a manual—this is in the supplementary but could be in the main text as well.

REPLY: Thank you, we have added a little more detail here.

Comment 10

Are microcosms open or closed to the atmosphere throughout the experiment? This should be made explicit, and a discussion of the amount of headspace would be helpful.

REPLY: Thank you, good point. Microcosms are closed except during the establishment of treatments and sampling. We have added the following text: "All microcosm incubators shall be closed after the filling procedure with the black screw cap (Fig. 1) and kept closed over the course of the experiment except during the establishment of treatments (section 2.5) and sampling. The enclosed headspace (Fig. 1) may vary slightly in between microcosms after the filling procedure (section 2.1) and will increase over the course of the experiment due to the withdrawal of samples. While an increasing headspace will lead to some limited $CO_2$ exchange between the atmosphere and the enclosed volume, previous studies with the same setting found that this has no effect on the OAE treatments established in the experiments (Guo et al., 2023; Ferderer et al., 2022)."

Comment 11

Line 218: 'Electrodialytical' could be simplified to 'electrochemical', as methods producing NaOH-based OAE may be based on electrodialysis or electrolysis.

REPLY: Thank you, agreed. Changed to electrochemical.

Comment 12

Line 228: On magnesium and calcium oxide based alkalinity sources—comparatively clean may depend on sourcing (industrial processes vs mining). It would be helpful to expand this comment—'comparatively clean to' (mined minerals? Olivines? In reference to heavy metals?) and to add citations throughout.

REPLY: Thank you, agreed. We have added more details according to the questions raised here and added references.

Comment 13

The intro (line 79) suggests 2 specific OAE scenarios, so I was expecting 2 alkalinity sources in section 2.2. Clarification in the intro and in this section would be helpful.

REPLY: Agreed and changed as detailed in our reply to comment 4.

Comment 14

Line 239—specify what longer is (decades vs centuries, for example). Other modeling studies could be cited here (e.g., Wang et al., 2023).

REPLY: Thank you, we specified the term "longer" as "potentially even centuries". We added (Wang et al., 2023) and (Mu et al., 2023) to the Jones et al reference.

Comment 15

Lines 247-259- discussion of the alkalinity addition chosen for this project is interesting. It would be very useful to place these estimates in context of the OAE studies that have been published and that are underway, which have ranged from 100-1000umol/kg.

REPLY: Thank you, we expanded this discussion and agree that a debate around this topic needs to happen within the OAE community. We note that there are so far only three published studies on how OAE effects plankton communities with fast dissolving alkalinity sources (or at least we could only find 3 in the literature). The perturbation levels were: 300-2400 µmol/kg (Paul et al., 2024); 150-600 µmol/kg (Ferderer et al., 2023); 500 µmol/kg (Ferderer et al., 2022). Even the lower values established here are arguably in the high end of what is realistic for OAE for the timescales of the experiments. For OAEPIIP we chose 500 µmol/kg to enable detectability of effects while still be in the extreme, yet somewhat plausible range. We have added a statement at the end of the section that future OAEPIIP outcomes will need to consider the degree of perturbation set up in the experiments (line 259-283).

Comment 11

Line 263- are microcosms allowed to temperature-equilibrate before the alkalinity addition?

REPLY: Temperature-equilibration will not be necessary because alkalinity is independent of temperature. The potential (minor) drifts in pH and pCO2 are considered acceptable.

Comment 12

Line 317- the section on seawater collection suggests gentle movements to avoid physical disturbance to plankton—how gentle is this manual stirring to break up Mg hydroxides?

REPLY: In our experiments the stirring was very gentle. The plastic paddle was rotated around once per 2-3 seconds. We reminded the experimentalist in the revised version that the microcosms should be stirred gently. To further limit the effect the protocol also describes that all microcosms need to be stirred equally and by an equal time.

Comment 13

Table 1/ S2—appreciate the detail on measured parameters. It would be useful to list out required accuracy/ precision of measurements. Table S2 calls out Dickson standards for DIC/TA/pH, but some users may assume glass electrodes are good enough for pH. Should call out scale in which pH should be reported. Would also be useful to state desired T/S/ etc precision—i.e., is a handheld salinometer good enough?

REPLY: Thanks, we agree with the sentiment and had quite long discussions about this within the team. Ultimately, we decided to not be overly prescriptive on minimum required precisions and accuracies. The reason is that each group will be better with some than with other parameters. Expecting everyone to reach perfection for each parameter would likely scare off potential participants and work against the capacity building component of OAEPIIP. We think that the treatment levels that will be established here are sufficiently high to iron out potential limitations in precision. (Having said that, we agree with the pH scale argument and added that pH shall be reported on the total scale.)

Comment 14

Line 368- provide an example of how short is too short and how long is too long.

REPLY: Thanks, we added an example.

Comment 15

Line 416: Description of models, GAMs, GAMMs would benefit from citations (i.e., bump up line 424).

REPLY: Thank you, now add the citations earlier as suggested.

Comment 16

Description of how specific parameters should be handled would be useful. Do you anticipate evaporation during these experiments, and if so, should carbonate parameters be normalized?

REPLY: Thank you, evaporation is a minor (negligible) problem when the experimental protocol is applied correctly (no normalization needed). We have added a description that microcosms are kept closed during the experiment, except for the short time of sampling and the establishment of treatment (Line 420-422).

Comment 17

Is there a reason dissolved oxygen is excluded from required parameters?

REPLY: There is no specific reason for that, other than the parameter does not usually lead to great insights in such microcosm experiments. Having said that, participants are free to measure O2 if they see additional value in it.

Comment 18

Line 450: are just the microcosms supplied, or also the heat belts, frames, lights, etc?

REPLY: Thanks for pointing that out. The funding for the microcosms, heat belts, lights, etc. is provided and will need be purchased by the participant. OAEPIIP helps with finding the right distributor in their respective regions. The microcosms used for OAEPIIP have global distribution. We clarified this in the revised version.

Comment 19

Does the $12000 include microcosms and publishing fees (line 450) or are those supplied additionally? If not, how much does that portion cost, i.e., how much does a participant have left over for analyses? Line 469 could be bumped up here to specify pub fees.

REPLY: Around 12,000 US$ is the total amount of funding provided. 1,600 US$ are reserved for publication fees. Thus, there are around 10,400 US$ available for materials and standardized components such as the microcosms. We clarified that in the revised version.

Comment 20

Lines 42, 293, 302, 378: add equation numbers.

REPLY: thanks, done.

Comment 21

Line 95: Suggest rearranging to call out figure 1 in the text before presenting it.

REPLY: thanks, we will address this issue for type-setting, depending on acceptance of the manuscript.

Reviewer #2

Comment 22

The OAEPIIP program proposed by Bach et al. would provide much-needed information about the potential environmental impacts of ocean alkalinity enhancement. The methodological approach they propose appears scientifically sound, but much of what they suggest in the way of experimental design has been published elsewhere (e.g., Ferderer et al., 2022; Iglesias-Rodríguez et al., 2023). Although Bach et al. do add valuable guidance for OAEPIIP participants – e.g., how to alkalinize seawater, recommendations for statistical

analysis – I don't find their overall methodological approach to be novel. The authors should also ensure that all relevant citations from Oschlies et al. (2023) are included in their manuscript. For example, Iglesias-Rodríguez et al. (2023) provide a list of recommended variables to be measured during OAE manipulations – including protocol references – that closely matches Table 1 of this manuscript; however, it was not cited.

Ferderer, A., Chase, Z., Kennedy, F., Schulz, K. G., and Bach, L. T.: Assessing the influence of ocean alkalinity enhancement on a coastal phytoplankton community, Biogeosciences Discussions, 2022, 1–36, https://doi.org/10.5194/bg-2022-17, 2022.

Iglesias-Rodríguez, M. D., Rickaby, R. E. M., Singh, A., and Gately, J. A.: Laboratory experiments in ocean alkalinity enhancement research, https://doi.org/10.5194/sp-2023-7, 2023.

Oschlies, A., Stevenson, A., Bach, L. T., Fennel, K., Rickaby, R. E., Satterfield, T., Webb, R., and Gattuso, J.-P.: Guide to best practices in ocean alkalinity enhancement research, 2023.

REPLY: We thank the reviewer for their kind words. The crucial component of OAEPIIP is not primarily the novelty of the experimental protocol but the global coordination, which requires a very detailed description of the methods.

Thanks for pointing out these references, which were indeed missing. We have added (Iglesias-Rodríguez et al., 2023), which also included a list of potentially relevant measurements for similar types of studies.

REFERENCES

Eisaman, M. D., Geilert, S., Renforth, P., Bastianini, L., Campbell, J., Dale, A. W., Foteinis, S., Grasse, P., Hawrot, O., Löscher, C. R., Rau, G. H., and Rønning, J.: Assessing the technical aspects of ocean-alkalinity-enhancement approaches, State of the Planet, 2-oae2023, 1–29, https://doi.org/10.5194/sp-2-oae2023-3-2023, 2023.

Ferderer, A., Chase, Z., Kennedy, F., Schulz, K. G., and Bach, L. T.: Assessing the influence of ocean alkalinity enhancement on a coastal phytoplankton community, Biogeosciences, 19, 5375–5399, https://doi.org/10.5194/bg-19-5375-2022, 2022.

Ferderer, A., Schulz, K. G., Riebesell, U., Baker, K. G., Chase, Z., and Bach, L. T.: Investigating the effect of silicate and calcium based ocean alkalinity enhancement on diatom silicification, Biogeosciences Discussions, 1–28, https://doi.org/10.5194/bg-2023-144, 2023.

Iglesias-Rodríguez, M. D., Rickaby, R. E. M., Singh, A., and Gately, J. A.: Laboratory experiments in ocean alkalinity enhancement research, State of the Planet, 2-oae2023, 1–18, https://doi.org/10.5194/sp-2-oae2023-5-2023, 2023.

Mu, L., Palter, J. B., and Wang, H.: Considerations for hypothetical carbon dioxide removal via alkalinity addition in the Amazon River watershed, Biogeosciences, 20, 1963–1977, https://doi.org/10.5194/bg-20-1963-2023, 2023.

Paul, A. J., Haunost, M., Goldenberg, S. U., Hartmann, J., Sánchez, N., Schneider, J., Suitner, N., and Riebesell, U.: Ocean alkalinity enhancement in an open ocean ecosystem: Biogeochemical responses and carbon storage durability, EGUsphere, 1–31, https://doi.org/10.5194/egusphere-2024-417, 2024.

Wang, H., Pilcher, D. J., Kearney, K. A., Cross, J. N., Shugart, O. M., Eisaman, M. D., and Carter, B. R.: Simulated Impact of Ocean Alkalinity Enhancement on Atmospheric CO2 Removal in the Bering Sea, Earth's Future, 11, e2022EF002816, https://doi.org/10.1029/2022EF002816, 2023.